# Isolation of Tetracycline-Resistant *Chlamydia suis* from a Pig Herd Affected by Reproductive Disorders and Conjunctivitis

**DOI:** 10.3390/antibiotics9040187

**Published:** 2020-04-17

**Authors:** Christine Unterweger, Lukas Schwarz, Martina Jelocnik, Nicole Borel, René Brunthaler, Aleksandra Inic-Kanada, Hanna Marti

**Affiliations:** 1Department for Farm Animals and Veterinary Public Health, University Clinic for Swine, University of Veterinary Medicine, 1210 Vienna, Austria; 2Genecology Research Centre, University of the Sunshine Coast, Sippy Downs, QLD 4556, Australia; 3Institute of Veterinary Pathology, Vetsuisse Faculty University Zurich, 8057 Zurich, Switzerland; 4Institute of Pathology, Department of Pathobiology, University of Veterinary Medicine, 1210 Vienna, Austria; 5Institute of Specific Prophylaxis and Tropical Medicine, Medical University of Vienna, 1090 Vienna, Austria

**Keywords:** *Chlamydia suis*, fertility problems, conjunctivitis, minimal inhibition concentration, multilocus sequence typing, recovery testing, tetracycline resistance

## Abstract

Due to various challenges in diagnosing chlamydiosis in pigs, antibiotic treatment is usually performed before any molecular or antibiotic susceptibility testing. This could increase the occurrence of tetracycline-resistant *Chlamydia (C.) suis* isolates in the affected pig population and potentiate the reoccurrence of clinical signs. Here, we present a case of an Austrian pig farm, where tetracycline resistant and sensitive *C. suis* isolates were isolated from four finishers with conjunctivitis. On herd-level, 10% of the finishers suffered from severe conjunctivitis and sows showed a high percentage of irregular return to estrus. Subsequent treatment of whole-herd using oxytetracycline led to a significant reduction of clinical signs. Retrospective antibiotic susceptibility testing revealed tetracycline resistance and decreased susceptibility to doxycycline in half of the ocular *C. suis* isolates, and all isolates were able to partially recover following a single-dose tetracycline treatment in vitro. These findings were later confirmed in vivo, when all former clinical signs recurred three months later. This case report raises awareness of tetracycline resistance in *C. suis* and emphasizes the importance of preventative selection of tetracycline resistant *C. suis* isolates.

## 1. Introduction

Chlamydial infections have been associated with a variety of diseases in pigs [1], including conjunctivitis [2,3], pneumonia [4], enteritis [5,6], and polyarthritis [7]. Additionally, *Chlamydia* spp. can cause a wide range of reproductive disorders such as abortions [8], perinatal mortality [9], vaginal discharge, repeated breeding [10,11], as well as poor reproductive performances in sows [12]. *Chlamydia (C.) suis* is the most prevalent chlamydial species in pigs [1,13]. The diagnosis of Chlamydiaceae infections, in particular, antibiotic susceptibility testing, is time-consuming and laborious due to their obligate intracellular nature. For diagnosing chlamydial infections in veterinary medicine, open questions containing the sampling type and timing, depend on the animal host species, clinical signs, or anatomical localization to test. Moreover, only few laboratories can offer the cultivation of these bacteria, restricting diagnosis to molecular methods, and thus any statement regarding their growth characteristics, virulence, and antibiotic resistance patterns will be missing [14]. Detecting evidence for the involvement of *C. suis* in the pathogenesis of fertility problems in sows is especially challenging. Therefore, *C. suis* infections are often diagnosed clinically following the exclusion of other well-known pathogens, but without the detection of *C. suis* using either nucleic acid amplification tests (NAATs) or serological methods, and without the identification of chlamydial inclusions using immunohistochemical or immunofluorescence staining [15]. Cultivation of these obligate intracellular bacteria is very laborious and expensive. Consequently, antibiotic susceptibility testing, which requires cell culture systems, is not performed on a routine basis.

Following the diagnosis of chlamydial infections in veterinary medicine, in the absence of an effective anti-*C. suis* vaccine, tetracyclines are usually the treatment of choice [15]. Tetracyclines are easy to apply via food or water. They are not on the WHO list of critically important antimicrobials [16] in contrast to macrolides, which are the treatment of choice in human chlamydial infections. In pigs, this could be a cause for concern considering that *C. suis* is the only chlamydial species known to have naturally acquired genes that encode tetracycline resistance [17,18]. Moreover, there is evidence for intra- and interspecies recombination upon co-infection in vitro such as *tetA*(C) transfer among *C. suis* isolates, as well as from *C. suis* to *C. trachomatis* [19,20]. All of these factors might have severe consequences for human health, considering that both *C. suis* and *C. trachomatis* DNA have been detected simultaneously in the eyes of trachoma patients in Nepal. Moreover, *C. suis* was isolated from ocular and rectal samples originating from slaughterhouse workers and pig farmers [20,21,22,23]. The use of sub-inhibitory concentrations of tetracycline, especially in the presence of tetracycline-resistant *C. suis* isolates, could lead to treatment failure and the selection of tetracycline resistance on herd-level with the potential recurrence of clinical signs [24,25]. However, despite the evidence for *C. suis* tetracycline resistance, the treatment of chlamydiosis in pigs is still limited to tetracyclines [15].

## 2. Case Study

The need for ethics approval in this case is deemed unnecessary according to national Austrian regulations (Tiergesundheitsdienstverordnung 2009, BGBl. II Nr. 434/2009), because data had been collected during routine diagnostic measures within the herd health management.

The case herd was located in Lower Austria in a family-owned farrow-to-finish farm housing 60 sows and 350 fattening pigs. In 2017, an increase of the irregular return to estrus rate over the last year, from 10% to more than 25% on average, was recorded. Sows of all parities were involved. About 20% of sows in estrus had a yellowish mucous vaginal discharge. Since then, oxytetracycline at an inconsistent concentration was fed during each estrus period for around five days, without any improvement. Abortions were not recorded. Conjunctivitis was not observed in sows, but in the fattening unit approximately 10% of the oldest finishers (19 and 22 weeks of age) showed severe reddening of the conjunctiva, prolapse of the third eyelid, and seromucous ocular discharge. No other clinical signs (e.g., fever, coughing, wasting, or diarrhea) were noted. Disinfection of the barns was not performed on a routine basis.

Due to the clinical signs, the veterinarian suspected *Chlamydia* spp. to be the causative agent of conjunctivitis in the finishers, which was confirmed by molecular and culture methods: in four out of five conjunctival swabs, a *Chlamydiales*-specific real-time PCR targeting a fragment of about 207–215 bp of the 16S rRNA region developed by Lienard [26], yielded positive results. Subsequent Sanger sequencing of purified PCR products (200 bp amplicon of the 16S gene) [26] was performed, and the sequences were compared against the National Center for Biotechnology Information (NCBI) database using BLAST-n, categorized according to the first 16S rRNA BLAST-hit identification, and the closest known organism found was *C. suis* (100% nucleotide identity). Furthermore, *C. suis* was successfully isolated from all four swabs following inoculation onto LLC-MK2 cells (rhesus monkey kidney cells). The species identity was then confirmed using established NAATs, a *Chlamydiaceae*-specific real-time PCR targeting a 111 bp sequence of the 23S rRNA [27], followed by species-identification using an Arraymate microarray [28,29], a method that can detect mixed infections with other chlamydiae [27,30]. Despite these findings, *C. suis* investigation of vaginal and cervical swabs taken from sows with vaginal discharge remained negative, but *C. suis* DNA was identified by PCR in the urogenital tract of one sow slaughtered due to irregular return to estrus [31], though isolation attempts remained unsuccessful.

Subsequently, the herd veterinarian started whole-herd treatment with oxytetracycline (40 mg/kg body weight/q 24 h) over 21 days, as in-feed medication. Additional improvements of biosecurity measures, mainly focusing on cleaning and disinfection, were put in place. Signs of conjunctivitis disappeared and fertility problems were reduced (less than 10 percent return to estrus rates, no vaginal discharge). Three months later, the farmer reported new cases of conjunctivitis in six pigs. At the same time, the fertility problems insidiously reoccurred. This time, fattening pigs were no longer treated, while sow treatment over the insemination time was continued.

Due to the recurrence of clinical signs, retrospectively, further investigations regarding the molecular characterization of the retained isolates taken at the first herd visit, prior to the antimicrobial treatment by *C. suis*-specific multilocus sequence typing (MLST) [32] and antimicrobial susceptibility testing [19], were performed (Table 1).

Following MLST, the isolates were denoted with new distinct sequence types (STs: ST 276 for isolate 330 MS, ST 277 for isolate 329 MS, ST278 for isolate 490 MS, and ST279 for isolate 494 MS), and the phylogenetic analyses clustered the isolates into two genetically distinct clades (Figure 1). The 494 MS isolate clustered in the first major clade together with other Swiss and US isolates [33,34], along with type strain S45, which was isolated in Austria in the 1960s [33]. The other three isolates clustered in the genetically diverse second major clade, which included other European, US, and Chinese *C. suis* pig isolates. The 490 MS, 329 MS, and 330 MS isolates also formed a distinct well-supported subclade. MS is the abbreviation for “fattening pig” in german.

For tetracycline and doxycycline susceptibility testing, all isolates were further tested for the presence of the *tet*A(C) resistance gene by a PCR assay established by Dugan et al. using the primer pair CS43/CS47 [17]. Two isolates (329 MS and 494 MS) were *tet*A(C)-positive, whereas the other two were (330 MS, 490 MS) negative. To verify the resistance of *C. suis* to tetracycline and doxycycline, the minimum inhibitory concentration (MIC) of tetracycline and doxycycline for all four isolates was determined as previously described [19]. The resulting consensus MIC was based on three values: First, the MIC value resulting from “initial phenotype” testing, which was based on a fast screening method developed by Marti et al. [19], where 96-well plates were simultaneously seeded and infected onto serial dilutions of the antibiotic of choice, and where the MIC was defined as the first concentration where the number of inclusions was strongly reduced compared to the control. Second, the value obtained through the method described by Suchland et al. [20], who defined the MIC as, “two times the concentration where over 90% of all inclusions were altered in size and morphology” compared to the control, and third, the MIC was determined according to Donati et al. [35], who defined the MIC as, “the lowest concentration that reduced the number of inclusions more than 90% compared to the level of drug-free controls” (Table 2a,b). A consensus MIC is necessary, because small discrepancies (2-fold differences) between assays are considered within the expected variations of these in vitro assays [19].

*Tet*A(C)-positive isolates (TcR), 329 MS and 494 MS, had high MIC values (4 µg/mL) against tetracycline (resistant if ≥4 µg/mL [35,36]) and were therefore confirmed to be resistant to tetracycline in vitro, while the MICs of *tet*A(C)-negative isolates (TcS), 330 MS and 490 MS, had MICs ranging from 0.06 to 0.125 µg/mL and were thus considered to be tetracycline sensitive. In contrast, no isolate showed in vitro resistance to doxycycline, although the MICs of TcR isolates were two- to eight-fold higher than those of TcS isolates (0.125–0.5 µg/mL compared to 0.06 µg/mL).

A recovery assay according to Marti et al. [18] was performed to determine the recovery from single-dose treatment with either tetracycline or doxycycline (Figure 2a,b). Briefly, all four isolates were exposed to low, moderate, and high concentrations of tetracycline (0.125, 0.5, and 2 µg/mL) or doxycycline (0.015, 0.06, and 0.25 µg/mL). After 48 h, the supernatant was either replaced with antibiotic-free (recovery, rec) or antibiotic-containing medium (continued exposure, exp), and incubated for another 48 h before samples were collected to infect fresh monolayers to measure the inclusion forming units per mL (IFU/mL).

The recovery assay confirmed the MIC results of tetracycline with markedly better recovery for the two TcR isolates (329 MS and 494 MS) compared to the TcS isolates (330 MS and 490 MS). Moreover, while doxycycline resistance could not be confirmed for certain isolates (MIC ≤ 4 µg/mL [37]), both MIC determination and the recovery assay indicate that the susceptibility of the TcR isolates was reduced compared to the TcS isolates.

## 3. Conclusions

We detected genetically diverse *C. suis* isolates in the herd described in this case report. Genetic diversity is consistent with previous studies [38,39,40], depicting the unprecedented diversity of the *C. suis* genome compared to other chlamydial species, which is strongly influenced by recombination and plasmid exchange. A broad diversity of isolates circulates within Europe, even within individual farms or within the same animal [39]. Together with studies from the USA, Switzerland, Japan, and China, this present case report on Austrian fattening pigs further illustrates a consistent diversity on a global level rather than regional clustering, even though *C. suis* is genetically quite diverse. This diversity has also been observed for other veterinary chlamydial species such as *C. pecorum* and *C. psittaci* [30,39,40].

While *C. suis* DNA was detected in the uterus of a slaughtered sow, fecal contamination during the slaughtering process could not be excluded. Difficulties in detecting *Chlamydia* in sows with reproductive failure are a common issue for veterinarians, who often decide to treat the sows with antibiotics regardless of the molecular findings, a strategy that was also employed in this case.

The ocular *C. suis* isolates were not only genetically different but also either resistant or sensitive to tetracyclines. Interestingly, both tetracycline sensitive and resistant isolates were isolated from animals that lived in the same pen, and were therefore in regular physical contact with each other during the study period, which coincides with the findings of a study investigating Swiss fattening pigs [36,39]. It could be hypothesized that resistance originates from the continued short-time application of oxytetracycline in sows at the time of insemination. The necessity of antimicrobial treatment as performed here, especially before valid susceptibility testing, is questionable, and should be viewed critically. However, it is performed in many herds where *Chlamydia* spp. are suspected to be involved in the fertility problems and are often the only possible choice due to the lack of anti-*C. suis* vaccines on the market.

A striking observation was the fact that both *tet*A(C)-positive isolates (329 MS and 494 MS) were tetracycline-resistant, but they only displayed decreased susceptibility to doxycycline in vitro compared to the two *tet*A(C)-negative, tetracycline and doxycycline sensitive isolates (330 MS and 490 MS). These findings stand in contrast to published results of Di Francesco et al., who concluded that the presence of a *tet*A(C)-containing genomic island is linked to both tetracycline and doxycycline resistance, but the findings are in line with another study from Germany [41]. The contrasting resistance in these two antibiotics could be explained by differences in terms of pharmacodynamics and pharmacokinetics between tetracyclines as a natural tetracycline, and doxycycline as a synthetic tetracycline [42]. The regular use of oxytetracycline, which is closer related to tetracycline than to doxycycline, in this herd over years could be a possible explanation for the differences in resistance between tetracycline and doxycycline. However, further studies are needed to investigate this phenomenon.

To estimate the resistance situation for *Chlamydia*, i.e., to confirm tetracycline resistance in *C. suis*, isolation and antibiotic susceptibility testing must be performed in addition to the detection of the *tet*A(C) gene by PCR, as there can be discrepancies between *tet*A(C) PCR results of clinical swab samples and in vitro testing for tetracycline resistance [36]. In this study, however, such discrepancies were not observed, giving a clear indication that the identified *tet*A(C) gene was part of *C. suis* and did not originate from another bacterial species.

As a final conclusion to this study, we propose to establish chlamydial cultivation as part of routine diagnostics in pigs for three reasons: First, infections with TcR *C. suis* isolates in the pork industry are rising [43,44,45,46], which could complicate the treatment of porcine chlamydiosis, and might even pose a threat for public health considering that transmission of *C. suis* to humans has been reported [36]. Second, TcR isolates cannot be conclusively identified without cultivation, because, while current molecular techniques may identify the *tet*A(C) gene, they cannot determine whether it originates from *C. suis* or another bacterial species. Finally, taking together the increasing number of TcR *C. suis* isolates and the inability to characterize them outside of in vitro assays, it is crucial to establish routine cultivation procedures in order to predict the clinical impact that TcR *C. suis* isolates may have on porcine health, a highly relevant question that has gained little attention so far, apart from case studies [44].

## Figures and Tables

**Figure 1 antibiotics-09-00187-f001:**
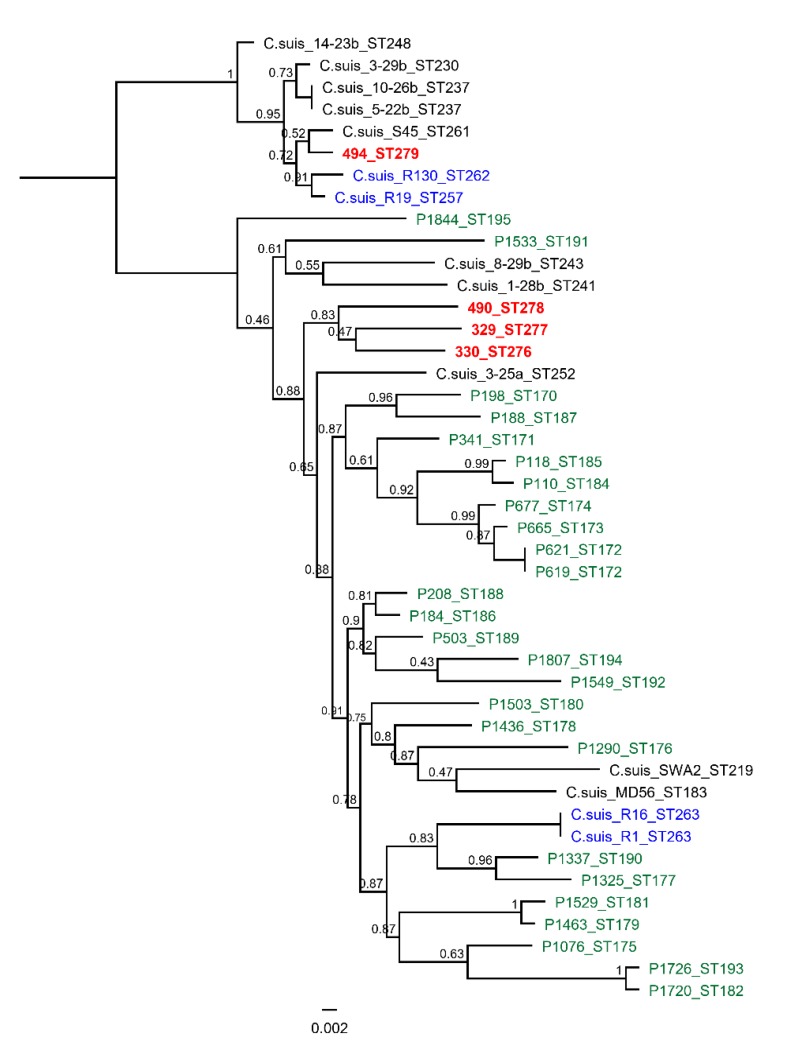
The mid-point rooted approximately-maximum-likelihood phylogenetic tree constructed using an alignment of concatenated multilocus sequence typing (MLST) sequences from the four isolates from this study, and an additional ten European, four US, and twenty-six Chinese *C. suis* isolates. The support values are displayed on the nodes. The isolates from this study are denoted in bold and red letters, other European strains in black, US strains in blue, and Chinese strains in green. ST for each is denoted at the end of the strain name. The figure was created in Geneious Prime v.2019.1, Biomatters (www.geneious.com).

**Figure 2 antibiotics-09-00187-f002:**
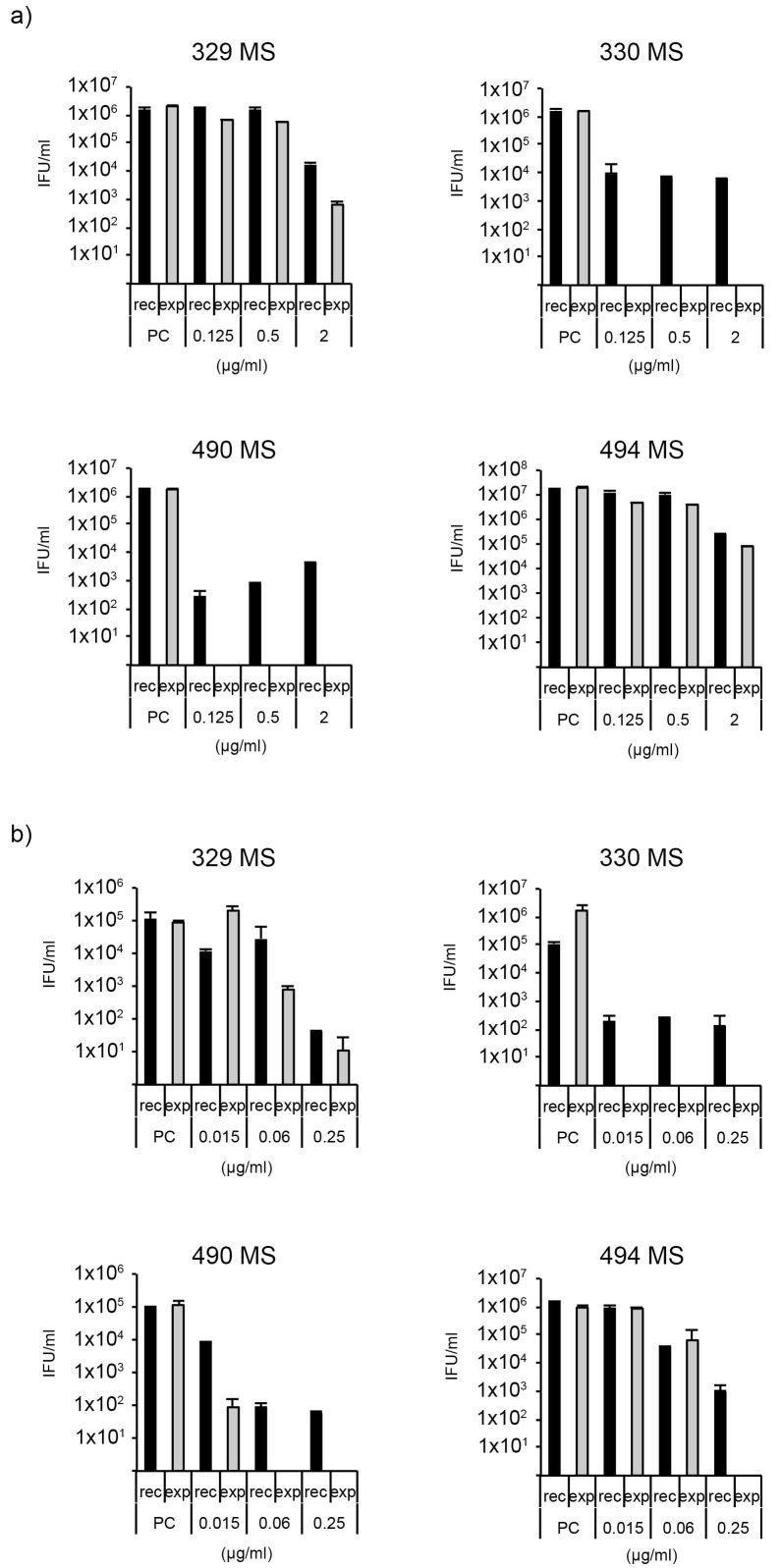
(**a**) Tetracycline recovery assay, and (**b**) doxycycline recovery assay according to Marti et al. [19]. X-axis: concentration of antimicrobials, y-axis: bars showing the number of viable elementary bodies EBs. Each recovery assay was performed once with two technical replicates. PC: positive control, rec: recovery group, exp: continuously exposed group.

**Table 1 antibiotics-09-00187-t001:** Summary of characteristics of ocular *C. suis* isolates 1–4.

	Isolate 1	Isolate 2	Isolate 3	Isolate 4
fattening ID	494 MS	490 MS	329 MS	330 MS
sequence type	ST279	ST278	ST277	ST276
phylogenetic clade	1. clade	2. clade	2. clade	2. clade

**Table 2 antibiotics-09-00187-t002:** (**a**) Susceptibility testing of tetracycline in vitro according to Marti et al. [19], and (**b**) susceptibility testing of doxycycline in vitro according to Marti et al. [19]. MIC: Minimum inhibitory concnetration.

(**a**)				
**MIC (µg/mL)**	**329 MS**	**330 MS**	**490 MS**	**494 MS**
Initial phenotype	2 to 4	0.125	0.125	4
MIC (Suchland)	4	0.06	0.125	4
MIC (Donati)	4	0.06	0.125	4
MIC (consensus)	4	0.06–0.125	0.125	4
Interpretation	resistant	sensitive	sensitive	Resistant
(**b**)				
**MIC (µg/mL)**	**329 MS**	**330 MS**	**490 MS**	**494 MS**
Initial phenotype	0.25	0.03–0.06	0.06	0.125
MIC (Suchland)	0.5	0.06	0.06	0.25
MIC (Donati)	0.25	0.06	0.06–0.125	0.25
MIC (consensus)	0.25–0.5	0.06	0.06	0.125–0.25
Interpretation	Reduced susceptibility	sensitive	sensitive	Reduced susceptibility

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
