# Peer review of "Isolation of Tetracycline-Resistant *Chlamydia suis* from a Pig Herd Affected by Reproductive Disorders and Conjunctivitis"

_antibiotics, 2020, doi:10.3390/antibiotics9040187_

Round 1
Reviewer 1 Report
The authors describe a case of an Austrian pig farm, where tetracycline resistant and sensitive Chlamydia suis strains were isolated from four finishers with conjunctivitis.
The tetracycline resistance in porcine C. suis strains is a potentially significant concern since it may not only affect the pig production, but also potentially play a role in public health.
Therefore, this study is of interest since it underlines the risk of tetracycline resistance (TetR) in C. suis due to the use of this antibiotic in pig herd that may contribute to trasmission of this pathogen to humans as well as to horizontal transfer of the TetR from C. suis to human clinical strains of C. trachomatis. Indeed, the strength of this study is the isolation of tetracycline-resistant C. suis strains from a pig herd since the cultivation of Chlamydiae is very difficult and laborious.
However, the manuscript has several issues that must be addressed. For example:
- The authors should better describe their experimental approach. The case herd included 350 fattening pigs and 60 sows; about 10% of the oldest finishers (19 and 22 weeks of age) showed clinical signs and 20% of sows in estrus had a yellowish mucous vaginal discharge. Why were conjunctival swabs collected only from 5 pigs?. Were new cases of conjunctivitis after antibiotic treatment investigated for the presence of C. suis? These findings may be helpful in the understanding the spread rate of antibiotic resistance among tetracycline sensitive C. suis strains and its impact on pig production;
- The authors may discuss more about the clinical significance of their results (initial susceptibility, MIC, recovery assay) and their impact on the surveillance and management of tetracycline resistance among pig population. In addition, the differences observed in the initial susceptibility test, MIC (Donati) and MIC (Suchland) regarding 329 MS and 330 MS strains should also discussed;
- The statement “To estimate the resistance situation for Chlamydia, e.g. to confirm tetracycline resistance in suis, isolation and antibiotic susceptibility testing must be performed in addition to the detection of the tetA(C) gene by PCR as there can be discrepancies between tetA(C) PCR results of clinical swab samples and in vitro testing for tetracycline resistance [38]” is not supported by their experimental data. Indeed, they found that TetA(C)-positive isolates were confirmed to be resistant to tetracycline in vitro as well as the TetA(C)-negative isolates resulted susceptible to tetracycline in vitro; TetA(C)-positive isolates only displayed decreased susceptibility to doxycycline in vitro. What are the relevant biological effects of these results in the context of C. suis infection in a pig herd? The pig herd was treated with oxytetracycline.Of note, the number of clinical strains analyzed in this study is too small in order to draw conclusion;
- the authors claim “Especially evidence for the involvement of suis in the pathogenesis of fertility problems in sows is challenging” Why were not C. suis investigated in sows with yellowish mucous vaginal discharge? The isolation of C. suis in sows with symptoms may be helpful to clarify the role of this microorganism in the pathogenesis of fertility problems;
- the statement “Nevertheless, chlamydial infections in pigs often remain undiagnosed and underreported, which happens mostly due to the lack of knowledge about a proper sampling type and timing, choice of diagnostic method as well as the interpretation of positive and negative results” should be more discussed.
-
Author Response
Response to reviewer 1:
Point 1: The authors should better describe their experimental approach. The case herd included 350 fattening pigs and 60 sows; about 10% of the oldest finishers (19 and 22 weeks of age) showed clinical signs and 20% of sows in estrus had a yellowish mucous vaginal discharge. Why were conjunctival swabs collected only from 5 pigs?Were new cases of conjunctivitis after antibiotic treatment investigated for the presence of suis? These findings may be helpful in the understanding the spread rate of antibiotic resistance among tetracycline sensitive C. suis strains and its impact on pig production;
Answer 1: This manuscript reports findings from a field study with samples taken during routine diagnostic investigation. Therefore, only a low number of conjunctival swabs was available to study. Unfortunately, a re-visit on this farm to take further samples from possibly re-infected animals was not feasible, although we fully agree that this information would have resulted in a more complete picture. Nevertheless, we could observe that clinical signs of conjunctivitis re-occurred and looked similar as described before.
Point 2: The authors may discuss more about the clinical significance of their results (initial susceptibility, MIC, recovery assay) and their impact on the surveillance and management of tetracycline resistance among pig population.
Answer 2: We thank the reviewer for this important input, however, as of today, systematic studies investigating the clinical significance of tetracycline-resistant compared to tetracycline-sensitive C. suis strains are missing, and any discussion related to this would be speculation. To date, one large-scale study found a correlation between high C. suis prevalence and the presence of diarrhea in fattening pigs (Hoffmann et al. 2015 PloS One), but no study has specifically looked at the effect of tetracycline resistance in this or another clinical context.
The reason for this knowledge gap relates to the fact that the identification of tetA(C) alone does not necessarily correlate with tetracycline resistant C. suis strains (Wanninger et al. 2016 Plos One) and that isolation of strains is very labor-intensive and costly, a drawback highlighted in the present study. In order to acknowledge this knowledge in the present study, the last paragraph (lines 218-228) was adjusted as follows:
"As a final conclusion of this study, we propose to establish chlamydial cultivation as part of routine diagnostics in pigs for three reasons. First, infections with TcR C. suis strains in the pork industry are rising [39,42–44], which could complicate the treatment of porcine chlamydiosis, and might even pose a threat for public health considering that transmission of C. suis to humans has been reported [38]. Second, TcR strains cannot be conclusively identified without strain isolation beause, while current molecular techniques may identify the tetA(C) gene, they cannot determine whether it originates from C. suis or another bacterial species. Finally, taking together the increasing number of TcR C. suis strains and the inability to characterize them outside of in vitro assays, it is crucial to establish routine cultivation procedures in order to predict the clinical impact that TcR C. suis strains may have on porcine health, a highly relevant question that has gained little attention so far apart from case studies [42]."
Point 3: In addition, the differences observed in the initial susceptibility test, MIC (Donati) and MIC (Suchland) regarding 329 MS and 330 MS strains should also discussed;
Answer 3: We thank the reviewer for this comment. However, the differences between these tests was never more than 2-fold, and is within the normal variation range that can be expected during in vitro assays as discussed in a previous study (Marti et al. 2018. Frontiers in Microbiology).
In order to make this clear, we added the following sentence to lines 143-145:
"A consensus MIC is necessary, because small discrepancies (2-fold differences) between assays are considered within the expected variations of these in vitro assays [18]."
Point 4: The statement “To estimate the resistance situation for Chlamydia, e.g. to confirm tetracycline resistance in suis, isolation and antibiotic susceptibility testing must be performed in addition to the detection of the tetA(C) gene by PCR as there can be discrepancies between tetA(C) PCR results of clinical swab samples and in vitro testing for tetracycline resistance [38]” is not supported by their experimental data. Indeed, they found that TetA(C)-positive isolates were confirmed to be resistant to tetracycline in vitroas well as the TetA(C)-negative isolates resulted susceptible to tetracycline in vitro; TetA(C)-positive isolates only displayed decreased susceptibility to doxycycline in vitro.
Answer 4: It is true that in this study, the PCR results correlated with the in vitro data, which is generally the case, but a more extensive study from the past showed that there can be discrepancies. However, we added the following note to the original statement (lines 216-218):
"In this study, however, such discrepancies were not observed giving clear indication that the identified tetA(C) gene was part of C. suis and did not originate from another bacterial species."
Point 5: What are the relevant biological effects of these results in the context of suis infection in a pig herd?
Answer 5: As mentioned above, the sample size was too small to draw any conclusions regarding the impact of tetracycline resistance in C. suis on porcine health.
Point 6: The authors claim “Especially evidence for the involvement of suisin the pathogenesis of fertility problems in sows is challenging” Why were not suis investigated in sows with yellowish mucous vaginal discharge? The isolation of C. suis in sows with symptoms may be helpful to clarify the role of this microorganism in the pathogenesis of fertility problems;
Answer 6: We agree with the Reviewer that these data are missing in the manuscript. We investigated sows with yellowish mucous discharge, but were not able to detect C. suis, neither by PCR nor by isolation. We even took sterile cervical swab samples and not only tested vaginal discharge. Standard bacteriological investigations on agar plates were performed and Streptococcus spp. (++) were found in two out of 10 samples, which is not unusual in cases of vaginal discharge. The negative test results for C. suis is not an exclusion of chlamydiosis in sows, it could also be due to the current lack of diagnostic knowledge (e.g. timing of sampling). Corresponding data were now added in the manuscript (lines 92-96).
Point 7: the statement “Nevertheless, chlamydial infections in pigs often remain undiagnosed and underreported, which happens mostly due to the lack of knowledge about a proper sampling type and timing, choice of diagnostic method as well as the interpretation of positive and negative results” should be more discussed.
Answer 7: This statement was extended in the introduction part (lines 40-46). However, a detailed discussion of the difficulties regarding the diagnosis and interpretation of porcine chlamydiosis would be too extensive and would go beyond the scope of this case report.
Reviewer 2 Report
Christine Unterweger et al. describes a case study about the appearance of Chlamydia suis tetracyclin resistant strains in a pig farm. The authors isolated C. suis from 4 infected animals. 2 of the isolates were proved to be tetracycline resistant, harboring the tetA(C) resistance gene and showed phenotypic signs of tetracycline resistance. The 2 other isolates did not harbor the resistance gene and were sensitive to tetracycline.
While the study is sound there are a few questions should be answered.
- The relationship between the isolated described by MLST typing may be discussed more. E.g isolates 490, 329, 330 are similar to each other (does it mean that they may have the same ancestor?), but one of them (490) is tetracycline resistant. The other tetracycline resistant isolate is 494 which is grouped to a different clade. Is it possible that the ancestor of 490, 329, 330 acquired the tetA(C) gene from 494?
- It is also worth to discuss that 330 and 490 had same/similar MIC values, but they clearly showed different capability for tetracycline recovery (1-1.5 log difference). It seems that 330 is more capable to go to persistence while tetracycline was more “cid” for 490.
- A novel finding was that the tetracycline resistance and the doxycycline were different, which is not in correlation with some of the earlier studies. What can be the cause of this phenomenon?
Altogether the described results are potentially acceptable for publication.
Author Response
Point 1: The relationship between the isolated described by MLST typing may be discussed more. E.g isolates 490, 329, 330 are similar to each other (does it mean that they may have the same ancestor?), but one of them (490) is tetracycline resistant. The other tetracycline resistant isolate is 494 which is grouped to a different clade. Is it possible that the ancestor of 490, 329, 330 acquired the tetA(C) gene from 494?
Response 1: While this is an excellent observation, the mechanism of tetA(C) transmission between C. suis strains is still under investigation. Recombination is suggested, but not necessarily proven. Moreover, the phylogenetic data in the present study is too limited to make such claims. First, whole-genome sequencing and analysis would be required and second, the general data set of C. suis genomic data would have to be vastly extended. Any statement regarding the ancestry of strains isolated and analysed during this study would remain purely speculative.
Point 2: It is also worth to discuss that 330 and 490 had same/similar MIC values, but they clearly showed different capability for tetracycline recovery (1-1.5 log difference). It seems that 330 is more capable to go to persistence while tetracycline was more “cid” for 490.
Response 2: We agree that the 1-1.5 log difference can be viewed as striking. However, the variability of chlamydial infectivity can be extremely high, especially between clinical isolates and following antibiotic treatment (Marti et al. 2018), and the differences between 330 and 490 would likely not be statistically significant if this assay was repeated several times. For this reason, we did not perform any quantitative analyses and focused on presenting our data qualitatively in order to clearly demonstrate differences in the recovery pattern of resistant and sensitive strains.
Point 3: A novel finding was that the tetracycline resistance and the doxycycline were different, which is not in correlation with some of the earlier studies. What can be the cause of this phenomenon?
Response 3: This is an excellent question that should be the focus of future studies that investigate the transcription and expression of tetA(C) and its repressor tetR(C) as well as structural analyses of the efflux protein comparing the two antimicrobial agents, doxycycline and tetracycline.
To further stress this out, we added the following paragraph to the discussion (lines 205 - 211):
"These contrasting resistance in these two antibiotics could be explained by differences in terms of pharmacodynamics and pharmacokinetics between tetracylines as a natural tetracycline and doxycycline as a synthetic tetracycline [41]. The regular use of oxytetracycline, which is closer related to tetracycline than to doxycycline, in this herd over years could be a possible explanation for the differences in resistance between tetracycline and doxycycline. However, further studies are needed to investigate this phenomenon."